# Substrate Diameter-Dependent Photovoltaic Performance of Flexible Fiber-Type Dye-Sensitized Solar Cells with TiO_2_ Nanoparticle/TiO_2_ Nanotube Array Photoanodes

**DOI:** 10.3390/nano10010013

**Published:** 2019-12-19

**Authors:** Bing-Chang Xiao, Lu-Yin Lin

**Affiliations:** Department of Chemical Engineering and Biotechnology, National Taipei University of Technology, 1 Sec. 3, Zhongxiao E. Rd., Taipei 10608, Taiwan; xerioe86507@gmail.com

**Keywords:** anodization, dye-sensitized solar cells, electrochemical impedance spectroscopy, flexible, fiber-type, TiO_2_

## Abstract

Fiber-type dye-sensitized solar cells (FDSSCs) are attractive as an energy source of soft electronics due to low-costs, non-toxicity and especially, their indoor-weak-light workable features. The TiO_2_ nanotube array (TNA) can grow on flexible Ti wires directly using anodization technique, which is convenient and can provide better contact between substrate/TiO_2_. However, a systematic study of assembling efficient TNA on photoanode of FDSSC is limited. This study investigated the anodization voltage and time effects of growing TNA on Ti wires. TiO_2_ nanoparticles (TNP) are fabricated on TNA using dip-coating technique to compensate for low dye adsorption of TNA. Dip-coating rate is varied to optimize TNP thicknesses to provide effective dye adsorption and charge-transfer routes. The highest photon-to-electricity conversion efficiency (*η*) of 3.31% was obtained for FDSSCs with TNA/TNP photoanode prepared using 60 V as the anodization voltage and 40 cm/min as the dip-coating rate. The influence of titanium wire diameter on *η* of FDSSCs was studied. The bending test was carried out on flexible FDSSC assembled using plastic tube. The photocurrent retention of 84% is achieved for flexible FDSSC bended for 10 times. This work firstly provides facile ways to assemble efficient photoanode with composite TiO_2_ structures for FDSSC and opens new insights on studying titanium wire natures on FDSSC performance.

## 1. Introduction

To achieve more comfortable life for human beings, soft electronics have attracted much attention nowadays [1,2,3]. The power supply of the soft electronics is indispensable for providing electricity to driving devices, such as sensors [4]. To fit the soft nature of the soft electronics, a flexible and stretchable power supply device is required. Moreover, the energy source needs to be easily acquired for widening the application of the soft electronics to unlimited places. In this regard, solar energy is the most suitable energy source for the power supply of soft electronics. Dye-sensitized solar cells (DSSCs) of low-cost, easy-fabrication and no toxicity are some of the promising power supplies for soft electronics [5,6,7]. Also, DSSCs can be driven by simply using the in-door weak light [8]. This advantage even broadens the application of DSSC as the power supply for soft electronics in the place without sunlight illumination. Therefore, the development of the flexible DSSCs, especially the flexible fiber-type DSSCs (FDSSCs), is indispensable for constructing efficient soft electronics.

Although every component of DSSCs plays certain visible roles on the photovoltaic performance, the photoanode with the dye-adsorbed nanocrystalline TiO_2_ has been considered the most significant part to converting photons to electrons [9,10,11]. Generally, the TiO_2_ nanostructure on the fiber-type photoanode of FDSSCs can be fabricated using two methods, including the deposition of TiO_2_ powder on the fiber-type conductive substrates, such as carbon fibers and metal wires [12] and the direct growth of TiO_2_ nanostructures on Ti wires [13,14,15,16,17]. The latter way as more often applied since the fabrication process is simpler and the contact between TiO_2_ and Ti wires would be better due to the direct growth technique. The TiO_2_ nanotube array (TNA) was proposed by Zwilling et al. in 1999 [18]. This work used chromic acid and hydrofluoric acid as the anodization electrolyte. Afterwards, the fluorine-contained electrolytes were widely used to carry out the anodization process and fabricate TNA on titanium substrate. Gong et al. reported the growth of TNA in a 0.5 wt % HF aqueous solution at room temperature using different anodizing voltages [19]. To increase the length of the TiO_2_ nanotube arrays, Ti was anodized with KF or NaF in the electrolyte [20]. Park et al. used NH_4_F to form TNA on the FTO glass [21]. As for the TNA application on DSSC, Liang et al. developed the flexible DSSC with the TiO_2_ nanotube arrays (TNA) photoanode, and achieved a photon-to-electricity conversion efficiency (*η*) of 5.1% and 6% decreases on *η* under 90° bending [13]. Pan et al. fabricated FDSSCs using the TNA photoanode and the carbon fiber counter electrode to attain an *η* value of 5.64% [14]. Fu et al. achieved an *η* value of 10% for a FDSSC composed of a hydrophobic/hydrophilic carbon nanotube counter electrode and a TNA photoanode. Liang et al. developed flexible FDSSCs with multiple TNA photoanodes, and achieved an *η* value of 6.6% for the FDSSC with six photoanodes [16]. Liang et al. obtained the TNA photoanode of flexible FDSSCs using two-step anodization and hydrothermal treatment. The FDDSSC presents an *η* value of 8.6% even subjected to 180° bending [17]. However, most of the previous works constructed the TNA photoanode and coupled the novel design of the carbon counter electrode to fabricate FDSSC. The detailed study on the growth of TNA and even the TiO_2_ composite with different morphologies is limited.

In this study, the highly ordered TNA was directly grown on the Ti wire using the anodization technique, and TiO_2_ nanoparticles (TNP) were in turn deposited on the TNA using the simple dip-coating method. The TNP/TNA composite structure was applied as the dye-adsorbed semiconductor on the photoanode of FDSSC. Different anodization voltages and times for assembling the TNA on Ti wires and different dip-coating rates for depositing TNP on TNA were applied to achieve high surface area for dye adsorption and suitable thickness of TNP/TNA for charge transfer. The DSSC with the optimized TNP/TNA photoanode presents an *η* value of 3.31%, while the DSSC with the pure TNA photoanode prepared at the same condition only shows an *η* value of 3.01%. It is expected that the TNA underlayer not only plays the role of one-dimensional charge-transfer path but also the light scattering layer to enlarge the light utilization, while the TNP overlayer is responsible for enhancing the dye adsorption and providing a higher possibility of light-to-electricity conversion. It is worth mentioning that the diameter of the Ti wire is 0.5 mm for our case, while the Ti wire with the diameter of 0.127 mm was used in most previous reports [14,22,23,24]. The FDSSC composed of the Ti wire with the diameter of 0.127 mm for its photoanode showed a smaller *η* value to the FDSSC with the photoanode prepared using the Ti wire with the diameter of 0.5 mm, owing to the thicker electrolyte layer for the former case to reduce the visible light absorption for the sensitizer. The plastic tube was also used to assemble the flexible FDSSC. The bending test was carried out in this work. The *J*_SC_ retention of 84% was attained when the flexible FDSSC was bent for 10 times. The novelty of this work lies on using the facile anodization and dip-coating methods to fabricate TNP/TNA composites for the flexible FDSSCs and on comparing the performance of FDSSCs with different diameters of Ti wires. However, the *η* value achieved in this work is smaller than those reported in the previous literature [13,14,15,16,17]. The most probable reason is the configuration of the cell assembly. Our system was constructed by using two parallel electrodes, but other configurations, such as the curved photoanode on the outside of the straight counter electrode, could be more favorable for light absorption and sensitizer adsorption. In the future work, different configurations of FDSSC will be assembled to understand the effects of the shape and the relative positions of photoanode and counter electrode on the photovoltaic performance of DSSC.

## 2. Experimental Section

### 2.1. Synthesis of TiO_2_ Nanotube Array and TiO_2_ Nanoparticle/Nanotube Array on Ti Wire

The Ti wires (diameters = 0.5 and 0.127 mm, ENVY, CO, USA) were sonicated successively in deionized water (DIW), detergent, acetone and isopropyl, and then rinsed with ethanol before drying. The TiO_2_ nanotube array (TNA) was fabricated on the Ti wire via anodization by using a power supply (MP-310, major science, Munich, Germany). Different voltages and times were used for anodization to optimize the growth of TNA on Ti wires. The electrolyte for anodization contained 0.3 wt % NH_4_F (98%, SIGMA-ALDRICH, St. Louis, MO, USA) and 8 wt % H_2_O in ethylene glycol (99%, SHOWA, Tokyo, Japan). The anodized Ti wire was rinsed using DIW and dried at room temperature. Subsequently, the anodized Ti wire was annealed in air at 500 °C for 1 h with the heating rate of 5 °C/min. Then the TNA coated Ti wire (TNA/Ti) electrode was thus obtained. Pictures of the Ti wire before and after anodization process are shown in Figure 1a,b. The white TiO_2_ layer was obviously obtained on the Ti wire after carrying out the anodization process. For the flexible photoanode of DSSC, the TNA-grown Ti wire is the most promising system. Since the bending condition is required for the flexible system, the attachment of TiO_2_ on the substrate should be paid extra attention. The TNA can grow on Ti wire by using the anodization process, and the attachment of TNA on Ti wire is very tight since the TNA is directly grown from the Ti wire substrate. Therefore, the method used in this work for assembling the photoanode is quite suitable for the flexible system. The TiO_2_ nanoparticle-coated TiO_2_ nanotube array (TNP/TNA) was fabricated on Ti wires as follows. The TNA was firstly fabricated on Ti wires by using anodization at 60 V for 6 h, and then TNP was coated on TNA/Ti by using the dip-coating technique. The TiO_2_ paste for dip-coating contains 6 g commercial TiO_2_ (P25), 2 mL of acetyl acetone and 0.1 mL Triton X-100 (VETEO) in 20 mL ethanol (99.5%, ECEO, Taipei, Taiwan). The thickness of the TNP layer was controlled by varying the withdrawing rates for dip-coating. Afterwards, the TNP/TNA/Ti electrode was sintered at 450 °C for 30 min with the heating rate of 5 °C/min. Then, the TNP/TNA/Ti electrode was obtained.

### 2.2. Assembly of Fiber-Type Dye-Sensitized Solar Cells

The fiber-type dye-sensitized solar cells (FDSSCs) were assembled using the dye-coated TiO_2_/Ti electrode as the photoanode and the Pt wire as the counter electrode. The as-prepared TiO_2_/Ti electrode was immersed into the sensitizer solution containing 0.3 mM N719 (99%, UniRegion. Bio-Tech, Taoyuan, Taiwan) and the mixture solvent of dehydrated acetonitrile/tert-butanol with a volume ratio of 1/1 for 24 h. The dye-coated TiO_2_/Ti photoanode and the Pt wire counter electrode were firstly put into a glass capillary or a plastic tube. The electrolyte was then injected into the tubular container, which was further sealed by using the hot melt adhesive. The assembly of the FDSSC was thus finished. The electrolyte for assembling the FDSSC contained 0.5 M lithium iodide (99%, ACROS, NJ, USA), 0.05 M iodine (98%, TCI, Tokyo, Japan), 0.5 M 1,2-dimethyl-3-propylimidazolium iodide (98%, TCI, Tokyo, Japan) and 0.5 M 4-tertbutyl-pyridine in dehydrated acetonitrile (99%, J.T.Baker, NJ, USA). The photo for the FDSSC was shown in Figure 1c.

### 2.3. Measurements and Characterizations

Morphology of TiO_2_ was observed using field-emission scanning electron microscopy (FE-SEM, Nova NanoSEM 230, FEI, Germany). Composition of TiO_2_ was examined using the X-ray diffraction patterns (XRD, X’Pert^3^ Powder, PANalytical, Netherlands). The photovoltaic performance of the FDSSC was evaluated using the photocurrent density-voltage (*J-V*) curve and the electrochemical impedance spectroscopy (EIS) technique, which were carried out by using potentiostat/galvanostat instrument with an FRA2 module (PGSTAT 204, Autolab, Eco–Chemie, Netherlands) under 1 sun (AM1.5) illumination with a solar simulator (X500, BLUE SKY, Taipei, Taiwan). The open-circuit potential and the frequency ranges from 0.01 Hz to 100 kHz were used to carry out the EIS measurement.

## 3. Results and Discussion

### 3.1. Morphology Characterization of TNA Electrodes and Photovoltaic Performances of FDSSCs

The TNA was grown on Ti wires by using the anodization technique. The anodization voltage and time for fabricating TNA were studied to understand the growth of TNA on the flexible Ti wire with the 360^o^ growing surface and to obtain the optimized condition for fabricating efficient TNA photoanode. The SEM images for the Ti wires anodized using 50, 60, 70 and 80 V for 6 h are shown in Figure 2a–l. Figure 2a–d presents the lowest magnified image with the whole Ti wire in the picture. It is clear to see that a layer of TNA was uniformly grown on the surface of the Ti wire. Also, there are some cracks between the TiO_2_ nanotubes especially for the case with the anodization voltage of 80 V. Figure 2e–h shows the larger magnification of the SEM image with obvious nanotube structures. Most of the nanotubes possess circular shapes at the open-end, but some of them present distorted circular shapes due to the crowded growth of the nanotubes during the anodization process. The diameters of 118, 135, 140 and 220 nm were, respectively, obtained for the nanotube anodized using 50, 60, 70 and 80 V. On the other hand, the length of nanotube was measured using Figure 2i–l. The nanotube lengths of 5.67, 10.07, 13.02 and 27.53 µm were, respectively, obtained for the nanotubes anodized using 50, 60, 70 and 80 V. The larger diameter and long tubes were attended for the sample prepared using larger voltage for anodization. This phenomenon is similar to those observed using the Ti foils as the substrate for anodization [25,26,27]. It was also found that the enhancements on the diameter and length of nanotubes increased largely when the anodization voltage of 80 V was applied for synthesizing TNA on the Ti wire. This phenomenon may be caused by the extremely high voltage of 80 V for anodization, which may overcome the solid surface and be able to etch the softer inner sides.

The photovoltaic performance of the FDSSCs with the TNA/Ti photoanodes was further evaluated by measuring the LSV curves, as shown in Figure 3. The photovoltaic parameters were listed in Table 1 for comparison. The open-circuit voltage (*V*_OC_) was decided at the zero current. Without current passing, the *V*_OC_ value is the difference between the conduction band edge of TiO_2_ and the redox potential of redox couple in the electrons. The charge recombination could reduce the *V*_OC_ value by decreasing the electrons in the conduction band of TiO_2_. The short-circuit current density (*J*_SC_) is determined at the short circuit condition, or the no voltage difference condition, which is the largest current achieved in the whole measuring process. The fill factor (*FF*) represents the ratio of the photovoltaic performance in the real conditions to that in the ideal conditions. In considering of the *V*_OC_, *J*_SC_ and *FF* values, the *η* value can represent the photon to electricity conversion efficiency for the DSSC [28,29]. The *η* values of 1.48%, 3.01%, 2.76% and 2.46% were respectively obtained for the DSSC with the photoanodes prepared using the anodization voltages of 50, 60, 70 and 80 V. The higher *η* value of the DSSC with the photoanode prepared using 60 V than that of the cell with the photoanode prepared using 50 V is owed to the higher *V*_OC_, *J*_SC_ and *FF* values for the former case, which is owed to the longer nanotubes with more dye molecules adsorbed to excite more electrons and the larger diameters of the nanotubes to allow more efficient electrolyte diffusion for carrying out more redox reactions for the TNA prepared using larger anodization voltage. However, the DSSC with the photoanode prepared using the anodization voltages of 70 and 80 V show smaller *η* values compared to that of the cell with the photoanode prepared using 60 V, even if the larger length and diameter of the nanotube can be synthesized by using larger anodization voltages. This result indicates that the larger lengths and diameters of the nanotubes are not definitely better as the photoanode of the DSSC. It was found that the *V*_OC_ value decreased for the DSSC with the photoanode prepared using larger anodization voltage except for the case prepared using 50 V. The decreased *V*_OC_ value suggests the more serious charge recombination, owing to the longer nanotubes with the length longer than the electron diffusion length. The *J*_SC_ values were also smaller for the DSSC with the photoanode prepared using 70 and 80 V, compared to that of the cell prepared using 60 V. Although the surface area of the nanotube may be larger for the TNA prepared using larger anodization voltages, the larger length of the nanotube may lead to serious charge recombination to reduce the charges successfully transferring to the outer circuit. Hence, the best photovoltaic performance was obtained for the FDSSC composed of the TNA/Ti photoanode prepared using 60 V as the anodization voltage, primarily due to the suitable length for efficient carrier transfer. The longer the TNA, the more dye molecules can adsorb on the surface of TiO_2_, but if the length of nanotube is too large the serious charge recombination could happen and the photovoltaic performance of the FDSSC could therefore be reduced, as observed for the FDSSC with the TNA/Ti photoanode prepared using 80 V as the anodization voltage.

In addition, based on the optimized anodization voltage of 60 V, different anodization times of 3 to 7 h were applied for growing TNA on Ti wires. The LSV curves for the FDSSCs with the TNA/Ti photoanodes prepared using different anodization times are shown in Figure 4. The photovoltaic parameters are listed in Table 2 for comparison. The FDSSC shows higher *V*_OC_, *J*_SC_ and *η* values when longer anodization times were applied for growing TNA on the Ti wire as the photoanode. The highest *η* of 2.68% was obtained along with the largest *V*_OC_ value of 0.77 V, *J*_SC_ value of 4.93 mA/cm^2^ and the *FF* of 0.71. However, when the anodization time of 7 h was applied for fabricating the TNA/Ti photoanode, the resulting FDSSC presented the worse performance with all the photovoltaic parameters reduced. This phenomenon may similarly be due to the length variation for the nanotubes anodized using different times. With increasing anodization times, the length of nanotubes would increase and hence the dye adsorption could be increased with more TiO_2_ grown on the Ti wire. However, the length of nanotubes may have exceeded the diffusion length of TiO_2_ when 7 h was used for anodization. The reduced photovoltaic performance of the FDSSC with the TNA/Ti photoanode anodized using 7 h was thus obtained.

### 3.2. Material Characterization of TNP/TNA Electrodes and Photovoltaic Performances of FDSSCs

Based on the optimized TNA electrode, the TNP was further deposited on the TNA using the dip-coating technique to increase the dye-adsorption amount for enhancing the electron excitation. Figure 5a–j shows the pure TNA and the TNP/TNA prepared using the withdrawing rates of 40, 20, 10 and 5 cm/min in the dip-coating process, respectively. The thickest TNP layer was obtained on TNA when the smaller withdrawing rate was applied for the deposition. The most cracks were also observed on the surface of the TNP layer prepared using smaller withdrawing rates due to the larger thickness. In addition, regardless of the thickness of the TNP layer, the nanoparticles are fully covered on TNA Figure 5g–j. Only slight portions of nanotubes were exposed at the surface, as found in Figure 5i. However, since the particle size of TNP was much smaller than the diameter of the nanotubes, it was inferred that the utilization of the surface area of nanotubes could not be sacrificed by the deposition of the TNP overlayer. Moreover, the composition of TNA and TNP/TNA was analyzed by using XRD patterns, as shown in Figure 6. The standard pattern of anatase TiO_2_ (JCPDS 84-1285) was also shown in this figure for comparison. All of the XRD patterns correspond to the standard pattern of anatase TiO_2_, indicating the anatase phase of nanotube prepared using anodization and the anatase phase of nanoparticles deposited on the nanotubes. The anatase TiO_2_ is preferable to the photoanode of DSSC, due to the suitable band gap and band positions as well as the high carrier mobility in this semiconductor. 

Furthermore, the photovoltaic performances of the FDSSCs with the TNP/TNA photoanodes were analyzed. Figure 7a shows the LSV curves of the FDSSCs with the TNP/TNA photoanodes prepared using the withdrawing rates of 5, 10, 20 and 40 cm/min in the dip-coating process. Table 3 lists the photovoltaic parameters for the FDSSCs with the TNP/TNA photoanodes prepared using different withdrawing rates in the dip-coating process. It was inferred that the thicker TNP layer could deposit on TNA when a smaller withdrawing rate was applied. The *η* value increases for the FDSSCs with the TNP/TNA photoanodes prepared using higher withdrawing rates. The highest *η* value of 3.31% along with the highest *J*_SC_ value of 5.95 mA/cm^2^ were obtained for the FDSSC with the photoanode prepared by using 40 cm/min as the withdrawing rate, probably owing to the thinnest TNP layer prohibiting serious charge recombination. However, the much thinner TNP layer is hard to obtain using the dip-coating technique, since the withdrawing rate of 40 cm/min is the smallest rate provided by the dip coater in our lab. The configuration of the TNP/TNA photoanode is beneficial for sensitizer adsorption and charge transportation. The larger surface area of TNP is advantageous for sensitizer adsorption and the vertically-grown TNA is preferable for electron transportation. Upon the light illumination, the charges will be exited from the sensitizer adsorbed on both TNP and TNA. The charges will transport from the TNP layer to the TNA underlayer, and the chargers in the TNA layer will then go to the conductive glass and to the outer circuit. Finally, the charges will arrive the counter electrode and participate in the redox reaction with the electrolyte. On the other hand, the charge transfer resistance was analyzed using the EIS technique. Figure 7b shows the Nyquist plot for the FDSSCs with the TNP/TNA photoanodes prepared using the withdrawing rates of 5, 10, 20 and 40 cm/min. The equivalent circuit is also shown in this figure for fitting the resistance data. The semicircle at the low frequency region can be an index of the charge transfer resistance at the interface between the photoanode and the electrolyte (*R*_ct2_) [30]. The *R*_ct2_ values of 217, 175, 140 and 125 Ω were, respectively, obtained for FDSSCs with the TNP/TNA photoanodes prepared using the withdrawing rates of 5, 10, 20 and 40 cm/min. The smallest *R*_ct2_ value for the FDSSC with the thinnest TNP layer on its photoanode indicates that the thickness of TiO_2_ layer plays an important role on the charge transferring. The smallest *R*_ct2_ value for the FDSSCs with the TNP/TNA photoanode prepared using the withdrawing rate of 40 cm/min was due to the thinnest TNP layer for inducing less charge recombination. This result is consistent with the best photovoltaic performance for this case.

### 3.3. Photovoltaic Analysis and Bending Test for FDSSC with Different Diameters of Ti Wire Substrates

To better understand the assembly of the FDSSC, different diameters of Ti wires were used as the substrate of the photoanode. The discussion in the previous text is based on the titanium wire with the diameter of 0.5 mm. In this section, the titanium wire with the diameter of 0.127 mm was used to fabricate the FDSSC. The TNA and TNP40/TNA were fabricated on titanium wires as the photoanode. The LSV curves for the FDSSC with the TNA and TNP40/TNA photoanodes prepared using the titanium wire with the diameter of 0.127 mm are shown in Figure 8a. Similar to the results obtained from the FDSSC prepared using the thicker Ti wire (0.5 mm) as the substrate, the FDSSC prepared using thinner Ti wire (0.127 mm) shows higher *V*_OC_, *J*_SC_ and *η* values with the TNP40/TNA photoanode than those for the cell with the TNA photoanode, owing to the more dye adsorption on the extra TNP layer for the former case. In addition, almost two fold higher *J*_SC_ and *η* values were obtained for the FDSSC prepared using the thicker titanium wire (0.5 mm) to assemble the photoanode, compared to those for the FDSSC with the photoanode prepared using the thinner titanium wire (0.127 mm). The smaller *J*_SC_ and *η* values for the FDSSC with the photoanode constructed using the thinner titanium wire are mainly due to the thicker electrolyte layer. The illustration of the glass tubes containing different diameters of Ti wires is shown in Figure 8b for more clear expression. Using the same size of the glass tube to assemble the FDSSC, the thinner titanium wire for the photoanode may leave more space for filling the electrolyte. The iodine in the electrolyte could absorb most parts of the visible light, so the thicker the electrolyte layer, the less visible light can be absorbed by the sensitizer on the TiO_2_. It was inferred that a thinner tube for assembling the FDSSC could lead to higher *J*_SC_ and *η* values due to the less light absorption of the electrolyte. However, the assembly of the FDSSC would be harder when the thinner tube is used as the container for placing the photoanode, counter electrode and the electrolyte. There is a trade-off for the ease of assembling the FDSSC and the photon-to-electricity conversion efficiency of the FDSSC due to the electrolyte issue.

On the other hand, the flexible FDSSC was also fabricated by using the plastic tube. The photo for the flexible FDSSC assembling using the plastic tube is shown as the insert of Figure 8c. To evaluate the stability of the FDSSC under the bending test, the LSV curve for the flexible FDSSC bent 10 times is also shown in this figure for comparison. Due to the higher light absorption of the plastic tube than that of the glass tube, the smaller *J*_SC_ value of around 1.21 mA/cm^2^ was obtained for the FDSSC assembled using the plastic tube. After bending 10 times, the *J*_SC_ value for the flexible FDSSC decreases to around 1.01 mA/cm^2^. The *J*_SC_ retention of around 84% was found for the FDSSC bended for 10 times compared with the value for the FDSSC without bending. To improve the stability under repeated bending, the sealing of the open-end for the tube container should be enhanced. Also, the flexibility of the plastic tube could be improved to decrease the light transmittance reduction after bending for several times. 

## 4. Conclusions

The TNP/TNA was fabricated on titanium wires as the photoanode of FDSSCs. The TNA was assembled on titanium wires using anodization techniques with different voltages and times. The optimized TNA was obtained using 60 V and 6 h as the anodization voltage and time, respectively. The FDSSC with the optimized TNA photoanode showed an *η* value of 3.01%, owing to the suitable diameter and length of nanotubes to provide sufficient dye-adsorption and charge-transfer path. The TNP was deposited on TNA using dip-coating technique with different withdrawing rates. The optimized thicknesses of TNP on TNA were obtained using 40 cm/min as the withdrawing rate, and the resulting FDSSC showed an *η* value of 3.31%. The photoanode fabricated using the smaller withdrawing rate may form a thicker TNP layer, whose thickness may exceed the diffusion length of TiO_2_ and increase the charge recombination possibility. Moreover, the FDSSC with the photoanode prepared using thinner titanium wire (diameter of 0.127 mm) shows smaller *J*_SC_ and *η* values than those of the FDSSC with the photoanode made using thicker titanium wire (diameter of 0.5 mm), owing to the thicker electrolyte layer for absorbing more visible light and reducing the light absorption of the sensitizer for the former case. The bending test was also carried out for the flexible FDSSC fabricated using the plastic tube. The *J_SC_* retention of 84% was achieved for the flexible FDSSC bent 10 times, compared to the value for the flexible FDSSC without bending.

## Figures and Tables

**Figure 1 nanomaterials-10-00013-f001:**
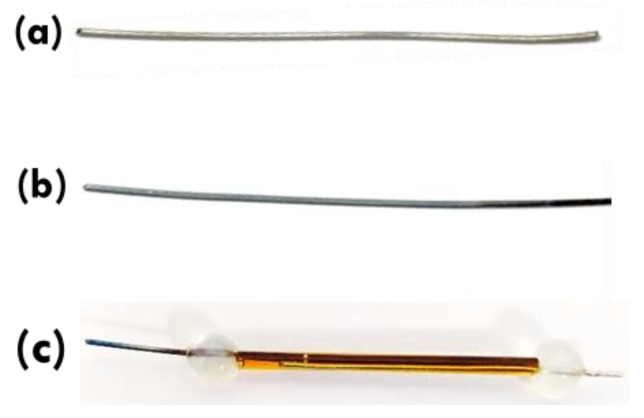
Pictures of (**a**) Ti wire, (**b**) anodized Ti wire and (**c**) fiber-type dye-sensitized solar cells (FDSSCs).

**Figure 2 nanomaterials-10-00013-f002:**
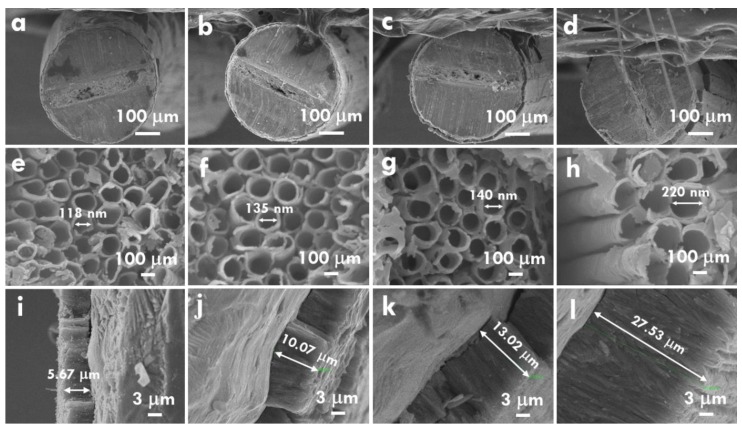
The SEM images for the TNA grown on Ti wire prepared using the anodization voltages of (**a**,**e**,**i**) 50, (**b**,**f**,**j**) 60, (**c**,**g**,**k**) 70, (**d**,**h**,**l**) 80 V.

**Figure 3 nanomaterials-10-00013-f003:**
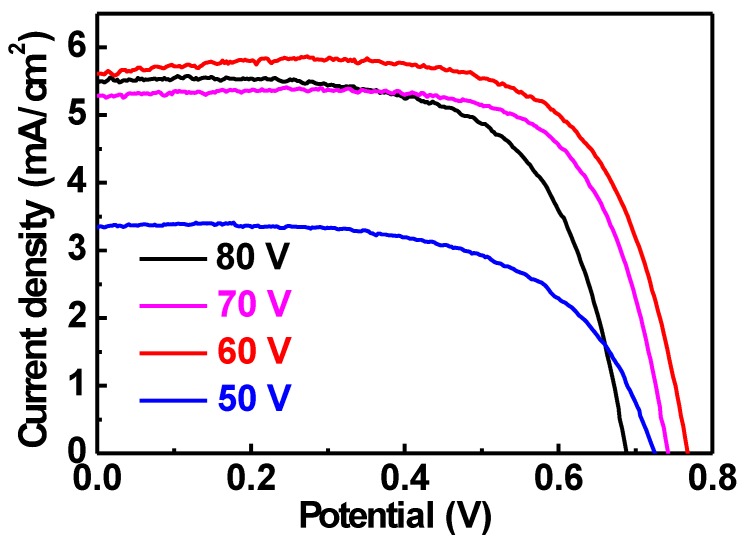
LSV curves for the FDSSC with the TNA/Ti photoanodes prepared using different voltages.

**Figure 4 nanomaterials-10-00013-f004:**
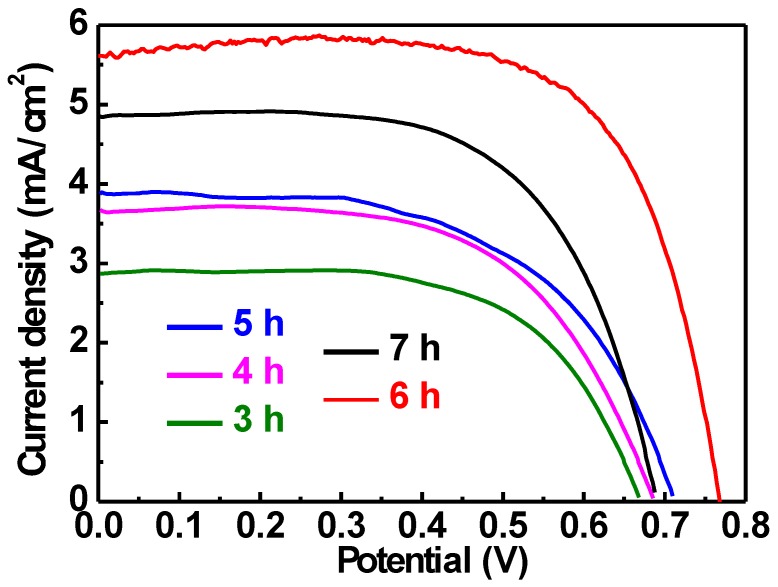
LSV curves for FDSSCs with TNA/Ti photoanodes prepared using different times at 60 V.

**Figure 5 nanomaterials-10-00013-f005:**
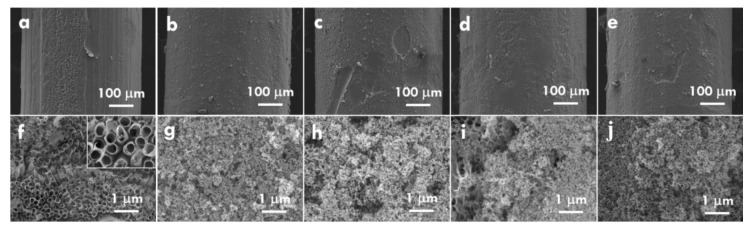
The SEM images for (**a**,**f**) the TNA and the TNP/TNA prepared using (**b**,**g**) 40, (**c**,**h**) 20, (**d**,**i**) 10 and (**e**,**j**) 5 cm/min as the withdrawing rates in the dip-coating process for depositing TNP.

**Figure 6 nanomaterials-10-00013-f006:**
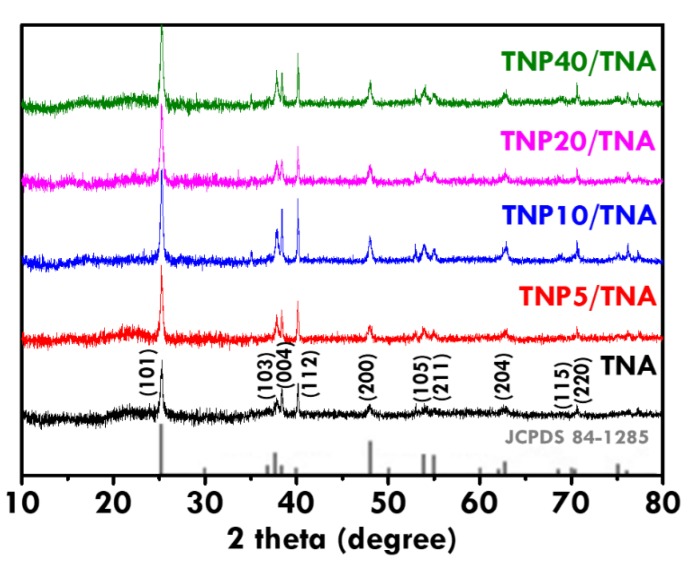
The XRD patterns for TNA and TNP/TNA prepared using different withdrawing rates.

**Figure 7 nanomaterials-10-00013-f007:**
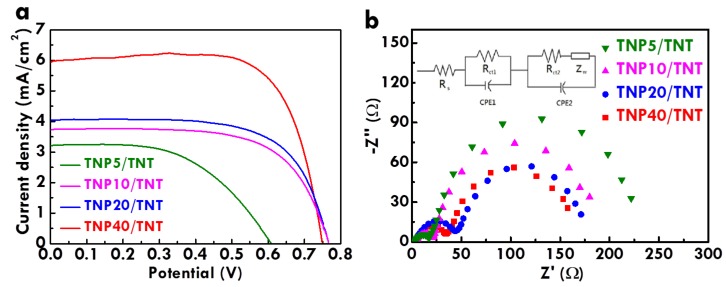
(**a**) The LSV curves and (**b**) the Nyquist plot for FDSSCs with the TNP/TNA photoanodes prepared using different withdrawing rates for depositing TNP in the dip-coating process.

**Figure 8 nanomaterials-10-00013-f008:**
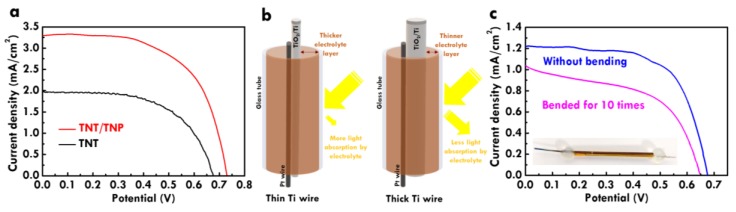
(**a**) LSV curves for the FDSSC with the TNA and TNP40/TNA photoanodes assembled using glass tube, (**b**) illustration of FDSSC with different diameters of Ti wires in the same size of glass tube and (**c**) LSV curves for FDSSC with TNP40/TNA photoanodes assembled using plastic tube without bending and with bending having been repeated 10 times. The diameter of Ti wire for measuring LSV curves was 0.127 mm.

**Table 1 nanomaterials-10-00013-t001:** The photovoltaic parameters for the FDSSCs with the TNA/Ti photoanodes prepared at different voltages over 6 h.

Voltage (V)	*V*_OC_ (V)	*J*_SC_ (mA cm^−2^)	*FF*	*η* (%)
50	0.73	3.35	0.60	1.48
60	0.77	5.61	0.70	3.01
70	0.74	5.29	0.74	2.76
80	0.69	5.49	0.65	2.46

**Table 2 nanomaterials-10-00013-t002:** The photovoltaic parameters for the DSSCs with the TNA/Ti photoanodes prepared using different times at 60 V.

Time (h)	*V*oc (V)	*J*sc (mA/cm^2^)	*FF*	*η* (%)
3	0.67	2.87	0.65	1.25
4	0.69	3.67	0.61	1.54
5	0.71	3.88	0.59	1.62
6	0.77	5.61	0.70	3.01
7	0.69	4.85	0.66	2.19

**Table 3 nanomaterials-10-00013-t003:** The photovoltaic parameters for the FDSSCs with the TNP/TNA photoanodes prepared using different withdrawing rates for depositing TNP in the dip-coating process.

Photoanode	*V*oc (V)	*J*sc (mA/cm^2^)	*FF*	*η* (%)
TNP40/TNA	0.75	5.95	0.74	3.31
TNP20/TNA	0.76	4.04	0.71	2.17
TNP10/TNA	0.77	3.73	0.67	1.93
TNP5/TNA	0.61	3.21	0.55	1.08

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
