# Peer review of "Substrate Diameter-Dependent Photovoltaic Performance of Flexible Fiber-Type Dye-Sensitized Solar Cells with TiO2 Nanoparticle/TiO2 Nanotube Array Photoanodes"

_nanomaterials, 2019, doi:10.3390/nano10010013_

Round 1
Reviewer 1 Report
The manuscript entitled "Substrate Diameter-dependent Photovoltaic Performance of Flexible Fiber-type Dye-sensitized Solar Cells with TiO2 Nanoparticle/TiO2 Nanotube Array Photoanodes" reports the development of TiO2 Nanoparticle/TiO2 Nanotube Array based system as an effective photoanode materials for the flexible fiber-type DSSC applications. The objective of the work is good. The designed material scheme and the obtained results are promising in the field. Therefore, this manuscript can be accepted for the publication. However, it requires the following moderate revision towards further improving the quality of the study conducted.
1. Provide the UV-visible absorption and PL spectra of the samples
2. Measure and report the band gap energy of the materials developed
3. Provide the photographic images of the Ti substrates (wires) before and after anodization process (Sec. 2.1.)
4. Also, provide the photographic images the fabricated DSSC device (Sec. 2.2)
5. Provide the general description of the various parameters such as VOC, JSC, FF, È , which will be useful for the general readers
6. Provide the incident photocurrent conversion efficiency graph of the developed DSSC (wavelength vs IPCE)
7. Discuss the mechanism of TiO2 Nanoparticle/TiO2 Nanotube Array system towards the observed enhanced properties towards DSSC applications
8. More references could be cited
Author Response
General comment
The manuscript entitled "Substrate Diameter-dependent Photovoltaic Performance of Flexible Fiber-type Dye-sensitized Solar Cells with TiO2 Nanoparticle/TiO2 Nanotube Array Photoanodes" reports the development of TiO2 Nanoparticle/TiO2 Nanotube Array based system as an effective photoanode materials for the flexible fiber-type DSSC applications. The objective of the work is good. The designed material scheme and the obtained results are promising in the field. Therefore, this manuscript can be accepted for the publication. However, it requires the following moderate revision towards further improving the quality of the study conducted.
Response to general comment: Thank the reviewer for the positive comments. We have modified our manuscript according to the suggestions from the reviewer thoroughly and carefully.
Specific comments
Comment 1: Provide the UV-visible absorption and PL spectra of the samples
Response to specific comment 1: Thank the reviewer for the suggestion. However, the photoanode is composed of Ti wire and TiO2 nanotubes. The Ti wire is opaque and the shape is wire-like, so it is hard and may be impossible to have the UV-visible absorption. The same reason was applied for measuring the PL spectra. Sorry for being unable to have UV-visible and PL spectra of the samples required by the reviewer. The UV-visible absorption may be required for examining the sensitizer absorption of light. We used the commercial N719 dye, so the light absorption spectrum is available on the website.
Comment 2: Measure and report the band gap energy of the materials developed.
Response to specific comment 2: Thank the reviewer for the suggestion. The band gap energy should be measured using the absorption spectra and the Tauc plot. However, due to the opaque nature of the photoanode composed of the Ti wire and the TiO2 nanotubes, the UV-vis spectra is impossible to obtain. Sorry for being unable to show the band gap energy of the material developed in this work.
Comment 3: Provide the photographic images of the Ti substrates (wires) before and after anodization process (Sec. 2.1.)
Response to specific comment 3: Thank the reviewer for the suggestion. The photos for the Ti substrates (wires) before and after anodization process were shown in Figure 1(a) and (b). The following sentences were added in lines 1-3 in the bottom at page 5 in the revised manuscript colored in the yellow background to explain the photos.
“The photos for the Ti wire before and after anodization process were respectively shown in Figure 1(a) and (b). The white TiO2 layer was obviously obtained on the Ti wire after carrying out the anodization process.”
Comment 4: Also, provide the photographic images the fabricated DSSC device (Sec. 2.2)
Response to specific comment 4: Thank the reviewer for the suggestion. The photo for the FDSSC was shown in Figure 1(c). The following sentences were added in the last two lines in the bottom at page 6 in the revised manuscript colored in the yellow background to explain the photo.
“The photo for the FDSSC was shown in Figure 1(c).”
Comment 5: Provide the general description of the various parameters such as VOC, JSC, FF, h, which will be useful for the general readers.
Response to specific comment 5: Thank the reviewer for the suggestion. The VOC value is decided at the zero current. Without current passing, the VOC value is the difference between the conduction band edge of TiO2 and the redox potential of redox couple in the electrons. The charge recombination could reduce the VOC value by decreasing the electrons in the conduction band of TiO2. The JSC value is determined at the short circuit condition, or the no voltage difference condition, which is the largest current achieved in the whole measuring process. The FF value represents the ratio of the photovoltaic performance in the real condition to that in the ideal condition. In considering of the VOC, JSC, and FF values, the h value can represent the photon to electricity conversion efficiency for the DSSC. The following sentences were added in lines 1-8 in the bottom at page 9 in the revised manuscript colored in the yellow background to explain the parameters.
“The open-circuit voltage (VOC) is decided at the zero current. Without current passing, the VOC value is the difference between the conduction band edge of TiO2 and the redox potential of redox couple in the electrons. The charge recombination could reduce the VOC value by decreasing the electrons in the conduction band of TiO2. The short-circuit current density (JSC) is determined at the short circuit condition, or the no voltage difference condition, which is the largest current achieved in the whole measuring process. The fill factor (FF) represents the ratio of the photovoltaic performance in the real condition to that in the ideal condition. In considering of the VOC, JSC, and FF values, the h value can represent the photon to electricity conversion efficiency for the DSSC [28-30].”
Comment 6: Provide the incident photocurrent conversion efficiency graph of the developed DSSC (wavelength vs IPCE)
Response to specific comment 6: Thank the reviewer for the suggestion. The IPCE value is not reliable to the fiber-shape system, since the total area of the fiber photoanode is much smaller than that for the normal-type plane photoanode. The intensity of the incident light at a single wavelength is not high enough to excite enough electrons for fair estimation. Therefore, we are sorry that the IPCE spectrum cannot be provided for this fiber-shape DSSC.
Comment 7: Discuss the mechanism of TiO2 Nanoparticle/TiO2 Nanotube Array system towards the observed enhanced properties towards DSSC applications
Response to specific comment 7: Thank the reviewer for the suggestion. The mechanism of TNP/TNA system towards the observed enhanced properties towards DSSC applications was discussed in the revised manuscript. The configuration of the TNP/TNA photoanode is beneficial for sensitizer adsorption and charge transportation. The larger surface area of TNP is advantageous for sensitizer adsorption and the vertical-grown TNA is preferable for electron transportation. Upon the light illumination, the charges will be exited from the sensitizer adsorbed on both TNP and TNA. The charges will transport from the TNP layer to the TNA underlayer, and the chargers in the TNA layer will then go to the conductive glass and to the outer circuit. Finally, the charges will arrive the counter electrode and participate in the redox reaction with the electrolyte. The following sentences were added from the last line at page 15 to the line 6 at page 16 in the revised manuscript colored in the yellow background to explain the mechanism.
“The configuration of the TNP/TNA photoanode is beneficial for sensitizer adsorption and charge transportation. The larger surface area of TNP is advantageous for sensitizer adsorption and the vertical-grown TNA is preferable for electron transportation. Upon the light illumination, the charges will be exited from the sensitizer adsorbed on both TNP and TNA. The charges will transport from the TNP layer to the TNA underlayer, and the chargers in the TNA layer will then go to the conductive glass and to the outer circuit. Finally, the charges will arrive the counter electrode and participate in the redox reaction with the electrolyte.”
Comment 8: More references could be cited.
Response to specific comment 8: Thank the reviewer for the suggestion. More references were cited in ref. 18-21 and 28-30 in the revised manuscript.

Reviewer 2 Report
Compare the efficiency obtained with the other similar technologies mentioned in the introduction and explain the reasons.
To analyze better the future prospects of utilization
Author Response
General comments: Compare the efficiency obtained with the other similar technologies mentioned in the introduction and explain the reasons. To analyze better the future prospects of utilization
Response to general comment: Thank the reviewer for the suggestion. The efficiency was compared, and the future prospects were provided in the revised manuscript. The following sentences were added in lines 4-10 at page 5 in the revised manuscript colored in the yellow background to explain the efficiency comparison and the future works. “However, the h value achieved in this work is smaller than those reported in the previous literatures [13-17]. The most possible reason is the configuration of the cell assembly. Our system is constructed by using two parallel electrodes, but other configurations such as the curved photoanode on the outside of the straight counter electrode could be more favorable for light absorption and sensitizer adsorption. In the future work, different configurations of FDSSC will be assembled to understand the effects of the shape and the relative positions of photoanode and counter electrode on the photovoltaic performance of DSSC.”

Reviewer 3 Report
The manuscript describes a new photo anode system for flexible DSSC.
Though the method described is interesting, a few concerns regarding the manuscript are
The present best values in flexible DSSC has not been met by this photoanode. Even if they have not crossed the current championship number, in what way does the method showcased in the manuscript play advantage over other methods in flexible electrode industry? These things have not come up in the current manuscript. The manuscripts has dealt with optimization and performance report of titanium nanotubes which have been well understood long back. Hence, they need to do an in-depth literature survey on titanium nanotubes. The uniqueness of the work is not highlighted properly.Author Response
General comments
The manuscript describes a new photoanode system for flexible DSSC. Though the method described is interesting, a few concerns regarding the manuscript.
Response to general comment: Thank the reviewer for the positive comment. We have modified our manuscript carefully according to the comments mentioned by the reviewer.
Specific comments
Comment 1: The present best values in flexible DSSC has not been met by this photoanode. Even if they have not crossed the current championship number, in what way does the method showcased in the manuscript play advantage over other methods in flexible electrode industry? These things have not come up in the current manuscript.
Response to specific comment 1: Thank the reviewer for the suggestion. For flexible electrode industry, especially for the flexible photoanode of DSSC, the TiO2 nanotube-grown Ti wire is the most promising system. Since the bending condition is required for the flexible system, the attachment of TiO2 on the substrate should be paid extra attention. The TiO2 nanotube can grow on Ti wire by using the anodization process, and the attachment of TiO2 nanotube on Ti wire is very tight since the TiO2 nanotube is directly grown from the Ti wire substrate. Therefore, the method used in this work to assemble the photoanode is quite suitable for the flexible system. Following sentences were added from the line 2 in the bottom at page 5 to the line 4 at page 6 in the revised manuscript colored in the yellow background to explain the advantage of the method used in this work.
“For the flexible photoanode of DSSC, the TNA-grown Ti wire is the most promising system. Since the bending condition is required for the flexible system, the attachment of TiO2 on the substrate should be paid extra attention. The TNA can grow on Ti wire by using the anodization process, and the attachment of TNA on Ti wire is very tight since the TNA is directly grown from the Ti wire substrate. Therefore, the method used in this work for assembling the photoanode is quite suitable for the flexible system.”
Comment 2: The manuscript has dealt with optimization and performance report of titanium nanotubes which have been well understood long back. Hence, they need to do an in-depth literature survey on titanium nanotubes. The uniqueness of the work is not highlighted properly.
Response to specific comment 2: Thank the reviewer for the suggestion. The literature survey on titania nanotubes was provided in the revised manuscript, and the following sentences were added in lines 14-20 at page 3 in the revised manuscript colored in the yellow background to explain the literatures.
“The TiO2 nanotube array (TNA) was proposed by Zwilling et al. in 1999 [18]. This work used chromic acid and hydrofluoric acid as the anodization electrolyte. Afterwards, the fluorine-contained electrolytes were widely used to carry out the anodization process and fabricate TNA on titanium substrate. Gong et al. reported the growth of TNA in a 0.5 wt% HF aqueous solution at room temperature using different anodizing voltages [19]. To increase the length of the TiO2 nanotube arrays, Ti was anodized with KF or NaF in the electrolyte [20]. Park et al. used NH4F to form TNA on the FTO glass [21]. As for the TNA application on DSSC,”

Round 2
Reviewer 1 Report
Authors have revised the manuscript satisfactorily and it can be accepted for the publication.
Author Response
Thanks the reviewer for the positive comment.